# Increase of Atmospheric Methane Observed from Space-Borne and Ground-Based Measurements

**Mingmin Zou [1], Xiaozhen Xiong [2,3,]\*, Zhaohua Wu [4], Shenshen Li [1], Ying Zhang [1] and Liangfu Chen [1]**

1   The State Key Laboratory of Remote Sensing Science, Institute of Remote Sensing and Digital Earth, Chinese Academy of Sciences, Beijing 100101, China; zoumm@radi.ac.cn (M.Z.); liss01@radi.ac.cn (S.L.); zhangying01@radi.ac.cn (Y.Z.); chenlf@radi.ac.cn (L.C.)
2   Science Systems and Applications, Inc., Lanham, MD 20706, USA
3   Earth System Science Interdisciplinary Center, University of Maryland, College Park, MD 20740, USA
4   Department of Meteorology & Center for Ocean-Atmospheric Prediction Studies, Florida State University, Tallahassee, FL 32306, USA; zwu@fsu.edu
\*   Correspondence: xiaozhen.xiong@noaa.gov; Tel.: +1-301-683-3606

**Abstract:** It has been found that the concentration of atmospheric methane ($CH_4$) has rapidly increased since 2007 after a decade of nearly constant concentration in the atmosphere. As an important greenhouse gas, such an increase could enhance the threat of global warming. To better quantify this increasing trend, a novel statistic method, i.e. the Ensemble Empirical Mode Decomposition (EEMD) method, was used to analyze the $CH_4$ trends from three different measurements: the mid–upper tropospheric $CH_4$ (MUT) from the space-borne measurements by the Atmospheric Infrared Sounder (AIRS), the $CH_4$ in the marine boundary layer (MBL) from NOAA ground-based in-situ measurements, and the column-averaged $CH_4$ in the atmosphere ($X_{CH4}$) from the ground-based up-looking Fourier Transform Spectrometers at Total Carbon Column Observing Network (TCCON) and the Network for the Detection of Atmospheric Composition Change (NDACC). Comparison of the $CH_4$ trends in the mid–upper troposphere, lower troposphere, and the column average from these three data sets shows that, overall, these trends agree well in capturing the abrupt $CH_4$ increase in 2007 (the first peak) and an even faster increase after 2013 (the second peak) over the globe. The increased rates of $CH_4$ in the MUT, as observed by AIRS, are overall smaller than $CH_4$ in MBL and the column-average $CH_4$. During 2009–2011, there was a dip in the increase rate for $CH_4$ in MBL, and the MUT-$CH_4$ increase rate was almost negligible in the mid-high latitude regions. The increase of the column-average $CH_4$ also reached the minimum during 2009–2011 accordingly, suggesting that the trends of $CH_4$ are not only impacted by the surface emission, however that they also may be impacted by other processes like transport and chemical reaction loss associated with [OH]. One advantage of the EEMD analysis is to derive the monthly rate and the results show that the frequency of the variability of $CH_4$ increase rates in the mid–high northern latitude regions is larger than those in the tropics and southern hemisphere.

**Keywords:** Methane increase trend; Boundary layer; Mid–upper troposphere; Satellite; AIRS

## 1. Introduction

As the third most important long life greenhouse gas after carbon dioxide ($CO_2$) and water vapor, atmospheric methane ($CH_4$) has a life time of about 12 years and its radiative forcing is about 26 times more than that of $CO_2$ on a 100-year time horizon, accounting for 32% of the total anthropogenic well-mixed greenhouse gas radiative forcing [1]. The concentration of $CH_4$ in the

atmosphere has increased from the pre-industrial levels of about 700 ppbv (parts per billion by volume) to 1800–1900 ppbv with a varying increase rate. Measurements from the ground-based networks operated by National Oceanic and Atmospheric Administration, Earth System Research Laboratory, Global Monitoring Division (NOAA/ESRL/GMD) since 1983 show that the increase rate of atmospheric $CH_4$ was less than 1 ppbv $yr^{-1}$ from 1999 to 2006 [2], nearly reaching a constant state, however, a rapid increase was observed since 2007. Dlugokencky et al. [3] estimated that $CH_4$ increased by $8.3 \pm 0.6$ ppbv in 2007, and $4.4 \pm 0.6$ ppbv in 2008, with the largest increase in the tropics. Analysis of preliminary data suggested a continued increase in 2014 [2]. A recent study by Bader et al. [4] showed an increase of atmospheric $CH_4$ total column of $0.31 \pm 0.03\%$ $year^{-1}$ using 10 years of in-situ measurements from 2005 to 2014, and based on GEOS-Chem model simulations, Bader et al. pointed out that anthropogenic emissions such as coal mining and gas and oil transport and exploration, which were mainly emitted in the Northern Hemisphere, had played a major role in the increase of atmospheric $CH_4$ observed since 2005. However, Nisbt et al. [5] concluded that fossil fuel emissions were not the dominant factor driving the recent increase and the major contributors were the increased tropical wetland and tropical agricultural methane emissions.

The most reliable measurements of atmospheric $CH_4$ are from the ground-based networks with flask sampling or in-situ measurements of the $CH_4$ mixing ratios in the marine boundary layer (MBL). Ground-based measurements also include the use of ground-based Fourier Transform Spectrometers (FTS) which provide bottom-up measurements of the total column amount of $CH_4$ using solar absorption spectra from two networks with long records: one is the Total Carbon Column Observing Network (TCCON), which currently has about 20 operational sites, and the other is the Network for the Detection of Atmospheric Composition Change (NDACC), which is composed of more than 70 high-quality, remote-sensing research stations. These ground-based measurements were used to validate space-based measurements [6–8]. In most cases, the NDACC sites also serve as TCCON sites and the only difference is the measurement of the spectral region. Aircraft measurements of $CH_4$ profiles were made by NOAA/ESRL/GMD as well as in some research campaigns [9]; however, they only provide sparse, intermittent measurements of $CH_4$ vertical profiles.

Overall, these ground-based measurements of $CH_4$ concentration in the MBL and aircraft measurements of $CH_4$ profiles are limited in time and space domain over the globe. Due to these limitations, the quantification of $CH_4$ emissions from different sources and/or in different regions still remains largely uncertain and the root-causes for the recent increase of $CH_4$ since 2007 are not quite clear. Therefore, the space-borne measurements from satellites, which have a better spatial and temporal coverage over the globe, would provide additional information to fill the gap among the ground-based measurements in time and space domains, regardless of the larger uncertainty as compared to in situ measurements. Up to present, there are over 10 years of observations from satellites. These measurements include the use of the thermal infrared (TIR) sensors, such as the Atmospheric InfraRed Sounder (AIRS) on NASA/Aqua [10–13], Tropospheric Emission Spectrometer (TES) on NASA/Aura [14–16], and more recent observations using the Infrared Atmospheric Sounding Interferometer (IASI) on METOP-A and -B [17–20]. Another type of space-borne measurement is the use of the Near-Infrared (NIR) sensors which include the SCanning Imaging Absorption spectroMeter for Atmospheric CHartographY (SCIAMACHY) instrument onboard ENVISAT from 2003 to 2009 [21,22], and the Thermal And Near infrared Sensor for carbon Observation (TANSO-FTS) onboard the Greenhouse gases Observation SATellite (GOSAT) from 2009 to present [23–26].

All these space-borne sensors can provide complementary observations to the variation of atmospheric $CH_4$ over the globe with a large spatial and temporal coverage [27]. AIRS has a product record of $CH_4$ from 2002 to present which has been well validated [11–13]. TES only has nadir and limb observations, so the samples of observations are very limited. IASI $CH_4$ data can be downloaded from NOAA, the Comprehensive Large Array-data Stewardship System (*CLASS*) [17], however it has not been reprocessed using the same version, thus the data is inconsistent due to some upgrade of algorithm in the past several years.

In this study, we use the space-born AIRS $CH_4$ products combined with the ground-based measurements from NDACC, TCCON, and NOAA/GMD to derive the changing trends of atmospheric $CH_4$ over the globe and at different altitudes. This trend analysis aims to give a better 3-D picture of $CH_4$ trends, thus helping us to better understand the increase trends of $CH_4$ during recent decades, especially since 2007. It is expected that these results will help the scientific communities to further explore the root-causes of the recent $CH_4$ increase, so as to better mitigate its potential impact on global warming. An adaptive time–frequency data analysis method, the Ensemble Empirical Mode Decomposition (EEMD), will be tested and applied to derive the $CH_4$ trend. A brief introduction of the three measurements' datasets and the EEMD method are described in Section 2. Section 3 presents the results with the discussion of the $CH_4$ monthly and annual increase rates in different regions from the three datasets. Finally, Section 4 is the conclusion.

## 2. Data and Method

### 2.1. Space-Borne Measurements from AIRS

As a thermal infrared sounder that has been very stable since its launch and of which is still under operation, AIRS on the EOS/Aqua satellite was launched in polar orbit (13:30 local standard time, ascending node) in May 2002. It has 2378 channels covering 649–1136, 1217–1613, and 2169–2674 cm$^{-1}$ at high spectral resolution ($\lambda/\Delta\lambda = 1200$) [8] and the noise in the equivalent change in temperature (Ne$\Delta$T) at a reference temperature of 250 K ranges from 0.14 K in the 4.2 μm in the lower tropospheric sounding wavelengths to 0.35 K in the 15 μm in the upper tropospheric sounding region. The spatial resolution of AIRS is 13.5 km at nadir and in a 24-hour period, AIRS nominally observes the complete globe twice. In order to retrieve $CH_4$ in both clear and partially cloudy scenes, nine AIRS fields-of-views (FOVs) within the footprint of the Advanced Microwave Sounding Unit (AMSU) are used to derive a single cloud-cleared radiance spectrum in a field-of-regard (FOR), which is then used to retrieve profiles with a spatial resolution of approximately 45 km. The atmospheric temperature profiles, water vapor profiles, surface temperatures, and surface emissivity are required as inputs to simulate the clear radiances in the $CH_4$ absorption band, and these inputs are retrieved from AIRS using different channels. The differences between the computed radiances and the AIRS measured radiances for clear pixels or the derived cloud-cleared FOR radiances for partially cloudy pixels are used to derive the change of $CH_4$ profiles. Then, 50–60 $CH_4$ absorption channels near 7.66 μm band were selected for the retrievals. The AIRS retrieval algorithm is a sequential retrieval method with multiple steps in which the temperature and water vapor profiles were retrieved using its own sensitive channels in previous steps before $CH_4$ retrieval, thus the quality of the $CH_4$ retrievals strongly depends on the AIRS retrieved temperature and moisture profiles as well as surface temperature and emissivity products. More detail of $CH_4$ retrievals in its most recent version, i.e. version six (V6), can be found in Xiong et al. [28]. The most sensitivity of AIRS to atmospheric $CH_4$ is in the mid–upper troposphere. The mid–upper tropospheric $CH_4$ dataset (AIRX3STD) retrieved using both AIRS and AMSU (Advanced Microwave Sounding Unit) observations are used in this study. Validation of the latest operational $CH_4$ product (AIRS version six) was made by comparing with aircraft data [28], which demonstrates that its RMS error (RMSE) is mostly less than 1.5%. Due to the failure of some AMSU channel, only the AIRS $CH_4$ data in the period from August 2002 to December of 2016 were used. From the global monthly averaged map of $CH_4$ at 400 hPa level in July 2003 and 2017 (Figure 1), we can clearly see the significant increase of $CH_4$ globally and the increase is up to 100 ppbv in most regions.

For comparison of $CH_4$ trends among the three measurements, a few regions of AIRS measurements were selected. Table 1 lists the latitudes and longitudes of the regions used. Daily $CH_4$ concentrations at the 400 hPa level are extracted from AIRS $CH_4$ profile products in each of these regions which are then used to calculate the regional average.

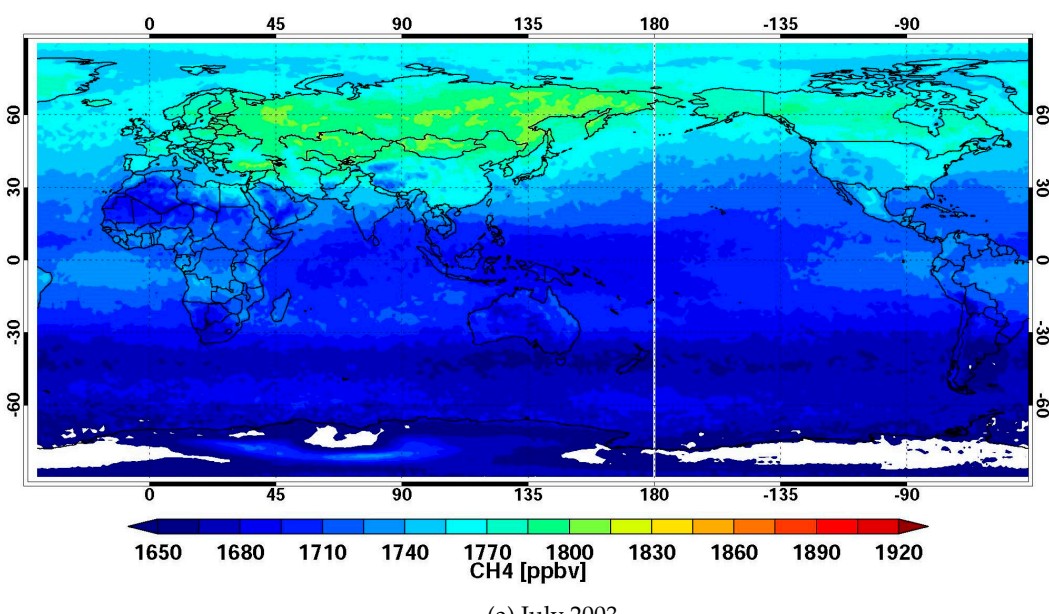

(a) July 2003

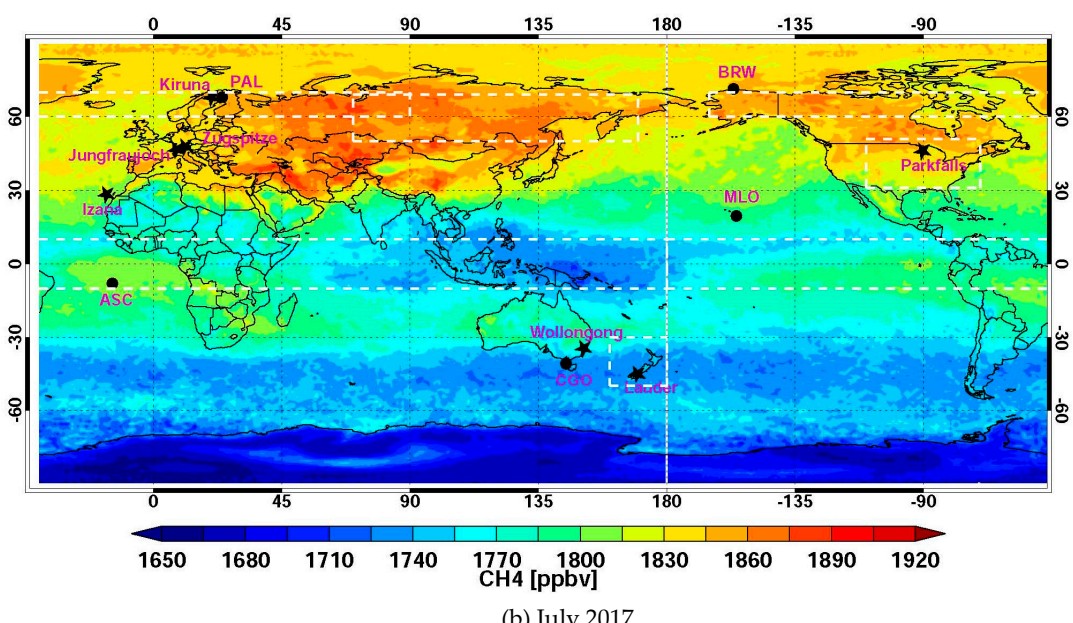

(b) July 2017

**Figure 1.** Global monthly mean $CH_4$ at 400 hPa level in July 2003 (**a**) and 2017 (**b**). Box areas indicated by dotted lines represent regions (listed in Table 1) used to calculate the Atmospheric Infrared Sounder (AIRS) average $CH_4$. Ground-based stations with surface $CH_4$ measurements used in this paper are marked with circles, as well as the ground sites with $CH_4$ total column measurements (listed in Table 2, Section 2.2) marked with stars.

**Table 1.** List of regions used to calculate the average of AIRS $CH_4$.

|  | Alaska-Canada (AK) | Siberia (SI) | Mid-Latitude (ML) | Tropics (TP) | S.Hemisphere (S.H) |
|---|---|---|---|---|---|
| Lat | 60 ~ 70 | 50 ~ 70 | 31 ~ 51 | −10 ~ 10 | −50 ~ −30 |
| Lon | −165 ~ 90 | 70 ~ 170 | −110 ~ −70 | −180 ~ 180 | 160 ~ 180 |

## 2.2. Column-Averaged $CH_4$ Measurements from the TCCON and NDACC Network

The Total Carbon Column Observing Network (TCCON) is designed to retrieve precise and accurate column abundances of $CO_2$, $CH_4$, $N_2O$, and CO, as well as HF, $H_2O$, and HDO from near infrared solar absorption spectra measured by ground-based Fourier Transform Spectrometers (FTS). The TCCON was established in 2004 [29] and currently there are 21 sites that are operational. The TCCON processed scheme is designed to minimize algorithmic biases between sites. To this end, a common and open source software package was used for data processing and analysis. The retrieval approach was carried out by the GFIT nonlinear least-squares fitting algorithm [30,31], which minimizes the residue between the recorded spectra and forward model calculation. It is estimated to achieve a precision of ~ 0.25% for $CO_2$ global monthly means [29,32] and ~ 0.5% for $CH_4$ tropospheric volume mixing ratios [33]. More information and data are available from https://tccon-wiki.caltech.edu/. TCCON $CH_4$ measurements acquired before and after AIRS pass time of ascending mode are selected to calculate the monthly mean value. Since the start date was not the same for different ground stations, we only selected two ground stations that have a longer period of measurements: one is Parkfalls located at the mid-latitude regions in the northern hemisphere [34–36] and the other is Lauder located in the mid-latitude of the southern hemisphere [37,38].

The international Network for the Detection of Atmosphere Composition Change (NDACC) is a major component of the international atmospheric research effort. The NDACC is composed of a set of globally distributed, high-quality, remote-sensing research stations for observing and understanding the physical and chemical state of the stratosphere and upper troposphere. Ground-based FTIRs from NDACC provide consistent, long-term measurements of the $CH_4$ column averaged volume-mixing ratio ($X_{ch4}$). We used $X_{ch4}$ from seven sites, which are listed in Table 2 to form the trend analysis.

**Table 2.** List of stations from Detection of Atmospheric Composition Change (NDACC), Total Carbon Column Observing Network (TCCON), and National Oceanic and Atmospheric Administration (NOAA)/ Global Monitoring Division (GMD) used for surface $CH_4$ trend analysis.

| Region | NO. | Station | Code | Latitude | Longitude |
|--------|-----|---------|------|----------|-----------|
| N.H | 1 | Barrow | BRW/NOAA | 71.32N | 156.61W |
|  | 2 | Pallas | PAL/NOAA | 67.97N | 24.12E |
|  | 3 | Kiruna | NDACC | 67.84N | 20.39E |
|  | 4 | Zugspitze | NDACC | 47.42N | 10.98E |
|  | 5 | Jungfraujoch | NDACC | 46.55N | 7.98E |
|  | 6 | Parkfalls | TCCON | 45.95N | 90.27W |
|  | 7 | Izana | NDACC | 28.29N | 16.48W |
| Tropics | 8 | Mauna Loa | MLO/NOAA | 19.54N | 155.58W |
|  | 9 | Ascension Island | ASC/NOAA | 7.97S | 14.40W |
| S.H | 10 | Wollongong | NDACC | 34.41S | 150.88E |
|  | 11 | Cape Grim | CGO/NOAA | 40.68S | 144.69E |
|  | 12 | Lauder | TCCON | 45.04S | 169.68E |

## 2.3. Surface $CH_4$ Measurements from the NOAA/ESRL/GMD Network

The NOAA Global Monitoring Division (GMD) is a world leader in producing the regional to global-scale, long-term measurement records which allow quantification of the most important drivers of climate change. Global monitoring of greenhouse gases has been part of NOAA's mission for over 50 years. NOAA/ESRL/GMD is the World Meteorological Organization (WMO), Global Atmosphere Watch (GAW) Central Calibration Laboratory (CCL) for $CO_2$, $CH_4$, $N_2O$, $SF_6$, and CO. The methane WMO scale was expanded to cover the nominal range 300 to 2600 ppbv based on the 16 original primary standards prepared in 1991–1995 [39]. The NOAA/ESRL/GMD network includes a cooperative program for the carbon gases providing air samples from more than 70 global air sites. Since these

ground measurements provide the most accurate measurements of $CH_4$, for comparison, datasets from ground-based in situ measurements of $CH_4$ in Barrow, Alaska (BRW) and Mauna Loa (MLO) were used for analysis. Another dataset from NOAA sampling network at Cape Grim, Tasmania (CGO) was also used for analyzing the trend in the Southern Hemisphere. These datasets are from ftp://aftp.cmdl.noaa.gov/data/trace_gases/CH4/flask/surface [40]. For further investigation of the surface $CH_4$ trend from the northern hemisphere to the southern hemisphere, another eight ground stations from NOAA/GMD were also used for analysis. Table 3 lists the stations that were used in this study.

*2.4. EEMD Method for Trend Analysis*

Ensemble Empirical Mode Decomposition (EEMD) is used to analyze the changing trend. EEMD is based on EMD [41,42], an adaptive time–frequency data analysis method that has been proved to be quite versatile in a broad range of applications for extracting signals from data generated in noisy nonlinear and nonstationary processes (see, e.g., [43,44]).

In EMD, the data *x(t)* are decomposed in terms of "intrinsic mode functions" (IMFs) $C_j$, and a residual component $R_n$, i.e.,

$$x(t) = \sum C_j(t) + R_n(t) = Re[\sum a_j(t)e^{i \int \omega_j(t)dt}] \tag{1}$$

In Equation (1), the residual component, $R_n$, could be a constant, a monotonic function, or a function that contains only a single extreme from which no more oscillatory IMFs can be extracted. The total number of IMFs of a data set is close to $\log_2 N$ [43,44], with *N* being the number of total data points. Different from many traditional decomposition methods, including the Fourier Transform and wavelet decomposition methods, EMD does not utilize a priori determined basis functions which may faithfully represent the characteristics of a time series in some segments, however not in other segments of a non-stationary time series. EEMD improves the stability of EMD when the extreme locations and values of analyzed data are changed by noise. More detail about EMD and EEMD can be found in Huang et al. and Wu et al. [43–45].

Temporal evolution of observables related to complex and not fully understood systems in nature is not a priori known. Using the EEMD derived trend of the obtained $_{CH4}$ concentration time series improves the non-physical choice of simple linear dependence in a completely new and logically consistent way [44]. Wu et al. [45] has made comparison of global-mean surface temperature trends derived using EEMD and a traditionally piece-wise linear fitting approach, respectively. The featured multi-decadal variability from the linear fitted trend cannot describe the secular warming, while the warming rate of the EEMD trend is changing gradually, which provides a chance to diagnose the warming rate changes of the secular trend. The uncertainty of the trend in the EEMD method is determined using a Monte Carlo approach rather than using an arbitrary disturbance such as white-noise. The details of this method can be found in the appendix of Wu et al. [45]. For any time series, its secular trend is supposed to be slowly varying and containing some noise.

As an example, to illustrate the EEMD used in this paper, Figure 2 shows AIRS time series of regional mean $CH_4$ at 400 hPa over Tropics and the EEMD decomposition results. The left panel of Figure 2 presents the residual components of decomposition. For a better view, $R_i$ is plotted by adding $(4 - i) \times 142$ (i = 1, 2, 3, 4); the original data (green line) and the final residua $R_4$ (red line) are plotted by adding $(4 \times 142 + 50)$. The right panel shows the IMFs of decomposition and, similarly, $C_i$ is plotted by adding $(4 - i) \times 20$ (i = 1, 2, 3, 4). It is found that the use of four IMFs is good enough to characterize the variation. The total number of IMFs, four, is close to the estimated number $\log_2 N$ in Equation 1, where N ≈14.1 years using AIRS data from August 30, 2002 to September 24, 2016. IMFs $C_1$, $C_2$, $C_3$, and $C_4$ represent the $CH_4$ variations with frequencies from high to low. The amplitudes of IMFs represent the strength of $CH_4$ variation. C1 shows a fast oscillation yet low amplitude. C2 and C3 show much stronger oscillations and these two components can reflect the seasonal behavior of $CH_4$. So, we will

combine C2 and C3 decomposed from three measurements to clearly present the seasonal behavior of $CH_4$ at different altitudes in Section 3.1. After three IMFS were extracted, C4 with the lowest oscillation frequency contain little information about $CH_4$ variation. The final residual $R_4$, represents the $CH_4$ trend from EEMD analysis. $R_4$ showed a rapid increase of $CH_4$ since 2007 and an even higher increase in later 2013, as observed from surface measurement. The annual increase rates are obtained as the derivative of $R_4$ with time.

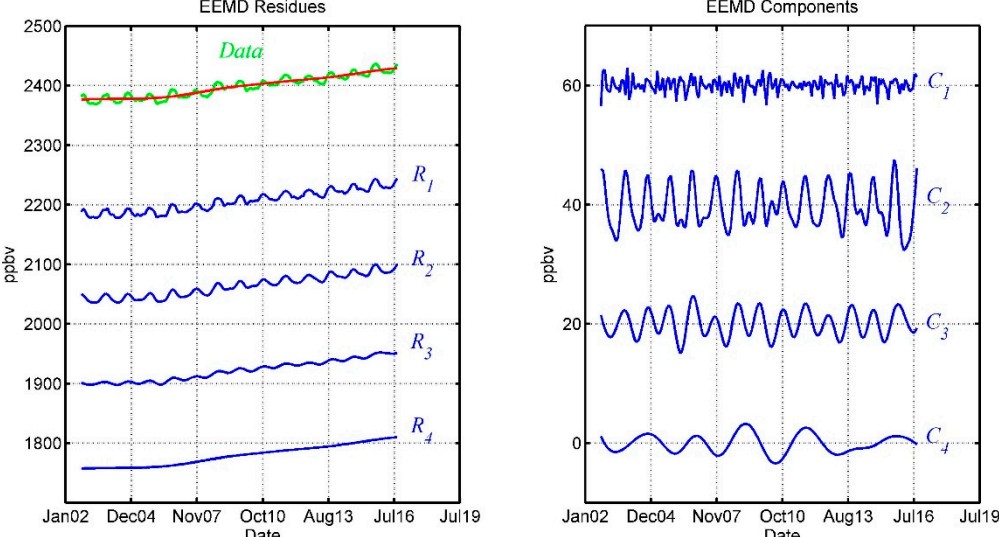

**Figure 2.** Ensemble Empirical Mode Decomposition (EEMD) decomposition of the AIRS $CH_4$ over the Tropics region. The left panel shows the original data (green line) and successive remainders ($R_j$) after an additional EEMD component ($C_j$) being extracted, i.e., $R_j = R_{j-1} - C_j$ for $j > 1$. The red line is the EEMD trend. In the right panel, each line represents an EEMD component ($C_j$) from high frequency to low frequency.

To further test EEMD used for trend analysis, data from six NDACC stations were analyzed using the EEMD decomposition. For each station, 10 years series of total column $CH_4$ ($X_{CH4}$) since 2005 were decomposed to get the residual R4. The $CH_4$ annual increase rates of these six stations were derived from R4 and the relative annual change percent was calculated referring to the monthly average in January 2005. Table 3 shows the relative changes of six stations and a comparison to those from Bader et al. [4]. We can see that the annual increases in all of these sites from this analysis using the EEMD method agrees quite well with those from Bader et al. [4] within the uncertainty range. One more advantage of the EEMD analysis is that we can derive the monthly rate of $CH_4$ increase as a function of time, from which we can get more information of the $CH_4$ change with time, making this research unique as compared to previous analysis.

**Table 3.** EEMD-based annual increase rates of $X_{ch4}$ at six NDACC stations during 2005–2014 and those from Bader et al.

| NO. | Station NDACC | Trend 2005~2014 (%/Yr) | Bader et al. 2005~2014 (%/Yr) |
|-----|---------------|------------------------|-------------------------------|
| 1 | Kiruna (67.97N) | 0.42 | 0.37 ± 0.04 |
| 2 | Zugspitze (47.42N) | 0.33 | 0.32 ± 0.03 |
| 3 | Jungfraujoch (46.55N) | 0.29 | 0.27 ± 0.03 |
| 4 | Izana (28.29N) | 0.32 | 0.33 ± 0.01 |
| 5 | Wollongong (34.41S) | 0.27 | 0.26 ± 0.02 |
| 6 | Lauder (45.04S) | 0.33 | 0.29 ± 0.03 |

## 3. Results and Discussion

### 3.1. Comparison of the Annual Increase Rates from Three Measurements

To be more concise for comparison, hereafter we will just plot the original data, four IMFs, and the residual $R_4$ in a single axis. Since all of these data are of different scales, plots will be made omitting the label of the y-axis. To investigate the difference of $CH_4$ variations between the mid–upper tropospheric (MUT) and surface, EEMD decomposition of AIRS $CH_4$ over the Alaska region at 400 hPa level was conducted. The IMFs and final residual R4 are presented at the left column of Figure 3. Also, the similar decompositions of surface $CH_4$ from BRW/NOAA station located near Alaska are shown in the right column for comparison. We combine C2 and C3 together to show a clearer seasonal cycle. According to Figure 3, it is evident that the IMFs amplitudes of surface $CH_4$ (BRW/NOAA) are larger than those from the AIRS region mean. This means that surface $CH_4$ has stronger seasonal variations. Besides, the residual R4 from two data sets both show a rapid increase since 2007, however the surface $CH_4$ has a much larger increase rate. The residue R4 from BRW/NOAA measurement also shows a pronounced rapid increase since later in 2013, while the residue R4 from AIRS shows an almost linear increase.

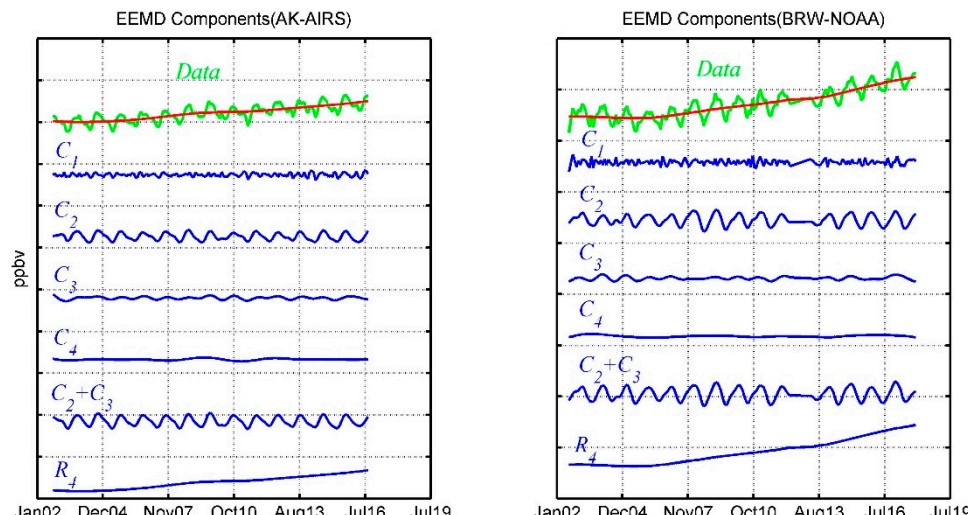

**Figure 3.** EEMD decomposition of the AIRS $CH_4$ over the Alaska region (left) and surface $CH_4$ (right) from the Barrow/NOAA/GMD station. The original data (green line) together with the EEMD trend (red line), four EEMD components ($Cj$), and the remainder $R_4$ are shown for each region.

We also make comparison between MUT and total column-average $CH_4$ ($X_{ch4}$) measurements. EEMD decompositions of AIRS $CH_4$ at 400 hPa level over the Mid-latitude region and $X_{CH4}$ from Parkfalls/TCCON site located in the Mid-latitude region are presented in Figure 4. Comparing the combinations of (C2 + C3), we can find a very similar seasonal cycle with almost the same variations from these two data sets. The residual R4 from both data sets also shows a rapid increase since 2007. Since observations from ground FTIR are more sensitive to surface emission than AIRS, R4 from total column-average $CH_4$ at Parkfalls/TCCON shows much larger increase rates than AIRS $CH_4$, especially after later 2013. This is consistent with the comparison between AK-AIRS and BRW-NOAA.

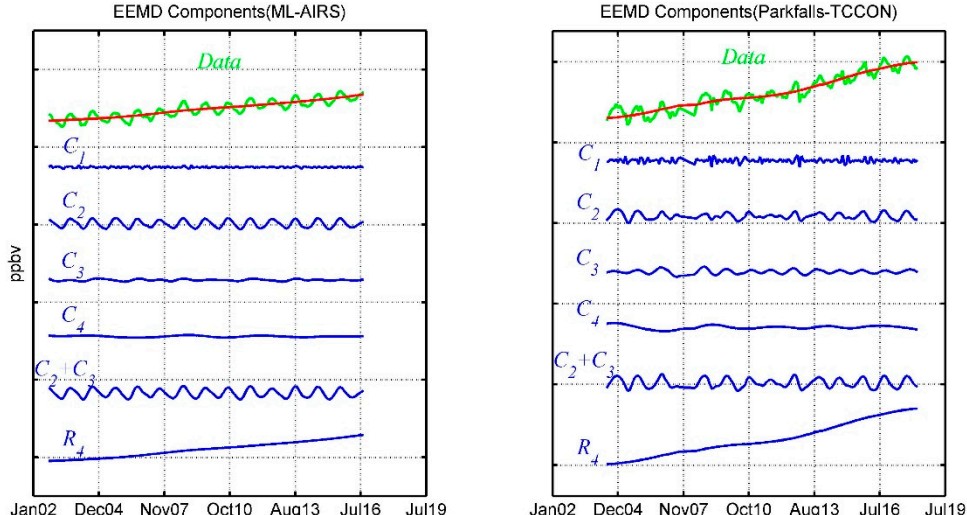

**Figure 4.** Same as Figure 3, however using AIRS $CH_4$ over the Mid-latitude region (**left**) and total column $CH_4$ ($X_{ch4}$) from Parkfalls/TCCON site (**right**).

To make further comparison for the three different data sets, we quantify the monthly increase rates from the derivation of EEMD-decomposed residual $R_4$ with respect to time. The annual increase rates are also calculated by integration of monthly rates. Figure 5 shows the comparison of $CH_4$ increase rates derived from AIRS and the ground-based stations in the northern hemisphere and Figure 6 shows the similar comparison in the tropics and southern hemisphere. In the northern hemisphere, we find that the three measurements agree well and capture the increase around 2007. The maximum of the column-average $X_{ch4}$ (Parkfalls/TCCON) occurred one year earlier than in the Arctic (Figure 5b,d). The second annual increase peaks of both $X_{ch4}$ and MUT $CH_4$ in the mid-latitude occurred in 2015. $CH_4$ in the MBL and in the MUT have similar pace when the maximum occurred (Figure 5a,c), however there is a slight phase shift (delay) of about three months over Siberia (February 2008) compared to Alaska (November 2007) and the maximum of MUT $CH_4$ over the Mid-latitude region occurred even earlier (August 2007). Things are different about the second increase peaks between Alaska and Siberia. Increase rates of both MUT and MBL $CH_4$ in Alaska reached the maximum in 2015, however the rate of MBL $CH_4$ in Siberia (PAL/NOAA) kept rising after 2015. During the period from 2009 to 2011, there are obvious dips in increase rates for the MUT, MBL, and total column $CH_4$ in the mid–high latitude regions. All the annual increase rates reach the minimum at year 2010. In the tropics and southern hemisphere (Figure 6), the phases of these three measurements agree quite well, however there is a delay of phase from the northern hemisphere to the tropics and a further delay in the high southern hemisphere. The shift of phases from MBL to MUT and from the northern hemisphere to southern hemisphere can be explained by the transport of $CH_4$ from the surface to the upper troposphere and transport from the northern hemisphere. The occurrence of the maxima of the column-average of $CH_4$ and AIRS $CH_4$ earlier than in other regions may indicate that the driver of $CH_4$ increase in 2007 is from the mid-latitude regions. This finding may echo a recent study by Schwietzke et al. [46] that found the emission of $CH_4$ from industrial sources is larger than previous estimates. The study by Bader et al. [4] also pointed out that anthropogenic emissions such as coal mining and gas and oil transport and exploration, which were mainly emitted in the Northern Hemisphere, had played a major role in the increase of atmospheric methane observed since 2005.

Vrekoussis et al. [47] pointed out that the economic crisis from 2008 resulted in a reduction of anthropogenic activities emitting gaseous pollutants, such as CO, $NO_2$, and $SO_2$. The strong correlation between the measured pollutants and several indicators of economic activity (Gross Domestic Product, Industrial Production Index, and Oil Consumption) may corroborate that economic recession has resulted in a reduction of air pollutants. Zhang et al. [48] also reported the reduction

of measured Volatile Organic Compounds (VOCs) in 2008 after the financial crisis compared to that in 2007, which is due to the reduced emission from industries that were hard-hit by the financial crisis. Amann et al. estimated the GHG mitigation potentials and costs based on the simulation from the GAINS (Greenhouse gas-Air Pollution Interactions and Synergies) model [49]. Two economic projections, WEO2008 and WEO2009, which were developed before and after the 2008 economic crisis, were adopted to evaluate the impact of the economic crisis. Simulations indicate that WEO 2009 projection of economic development led to 8% less GHG emissions for 2020 compared to WEO 2008. So, it is likely that the financial crisis in 2008 led to the slow-down of the $CH_4$ increase rate until 2010. Bergamaschi et al. [50] also pointed out that a large increase in anthropogenic $CH_4$ emissions first occured from 2006 and beyond.

It is noteworthy that the increase rates reversed later and reached another peak in 2014 which was even larger than in 2007. In this cycle in 2014, the peak in mid-latitude from TCCON took the lead and the peaks in the tropics and southern hemisphere still lag behind the northern hemisphere. This difference suggests that the emissions from human activities in the mid-latitude is likely playing a major role, which is also echoed from the slow-down of $CH_4$ in 2009–2010 corresponding to the financial crisis in 2008.

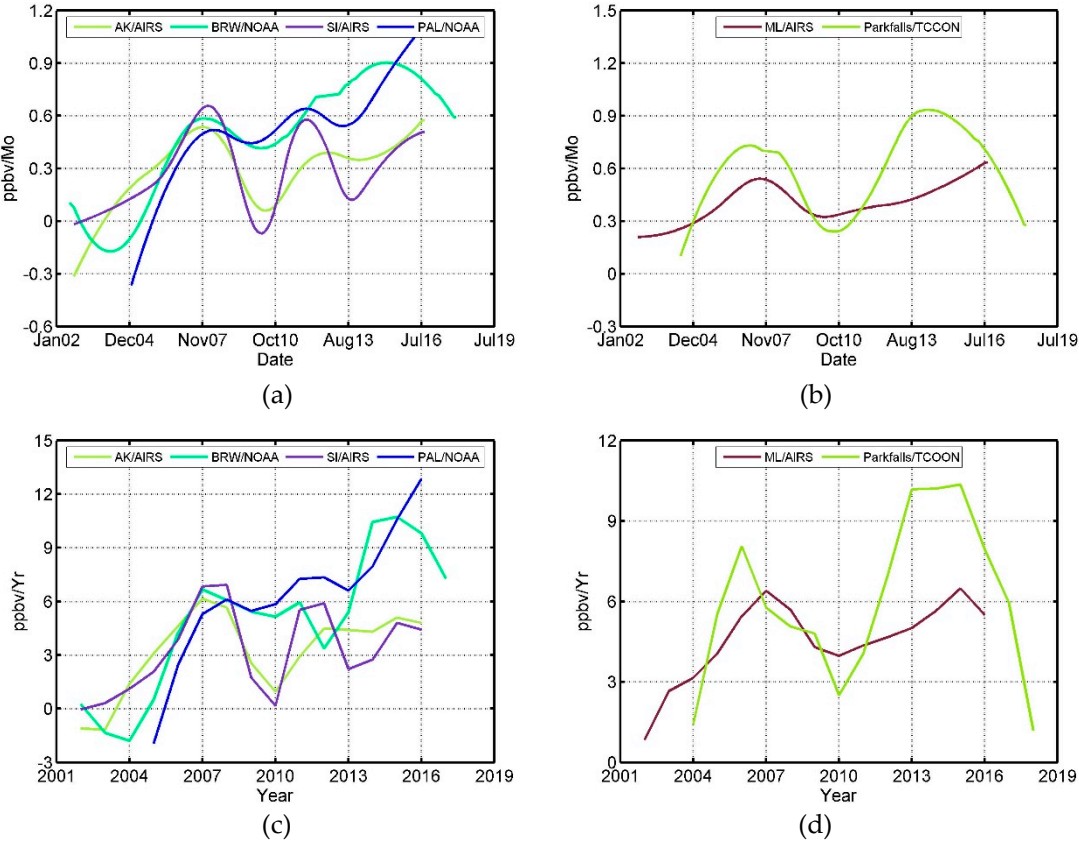

**Figure 5.** Comparisons of monthly and annual $CH_4$ increase rates (ppbv per month, ppbv/Mo and ppbv per year, ppbv/Yr) in the Northern Hemisphere from AIRS, TCCON, and NOAA ground-based measurements. (**a**,**c**) show monthly and annual AIRS $_{CH4}$ increase rates at 400 hPa over Alaska and Siberia regions and also surface $CH_4$ increase rates from two NOAA ground stations located in these two regions, respectively; (**b**,**d**) show monthly and AIRS $CH_4$ increase rates over the Mid-latitude region defined above and $X_{CH4}$ increase rates from Parkfalls/TCCON.

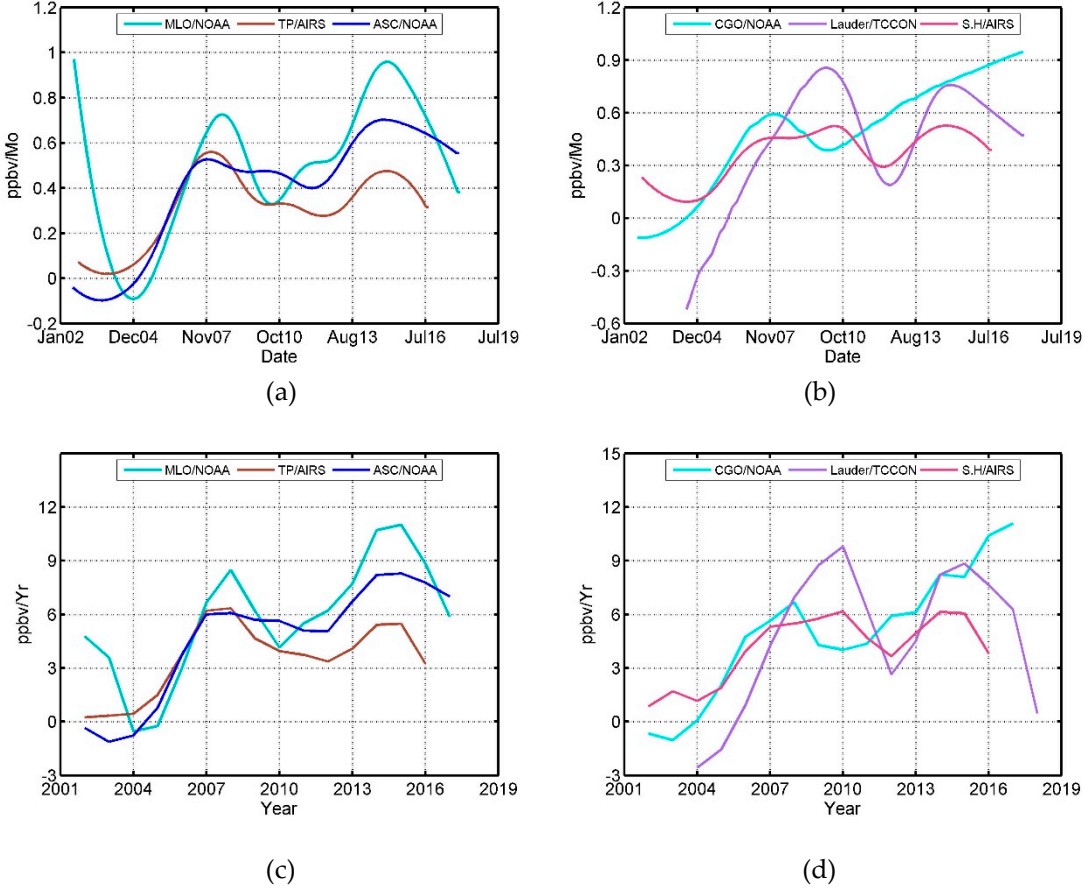

**Figure 6.** Comparisons of monthly and annual CH$_4$ increase rates (ppbv per month, ppbv/Mo and ppbv per year, ppbv/Yr) in the Tropics and Southern Hemisphere from AIRS, TCCON, and NOAA ground-based measurements. (**a,c**) show monthly and annual AIRS CH$_4$ increase at 400 hPa over the Tropics (TP) region and surface CH$_4$ increase rates from two tropical NOAA stations; (**b,d**) shows AIRS CH$_4$ increase rates over the southern Hemisphere, surface CH$_4$ from CGO/NOAA, and X$_{CH4}$ from Lauder/TCCON.

### 3.2. Trend of Zonal Mean CH$_4$ in the Mid–Upper Troposphere

To have a better picture of the global CH$_4$ variation with regards to time, EEMD decomposition can be applied to AIRS zonal mean CH$_4$ at 400 hPa level for trend analysis. Global zonal mean CH$_4$ is calculated at monthly scale and its time series from 2003 to 2016 is shown in Figure 7a. It is clearly shown that the mean CH$_4$ is much higher in the northern hemisphere than in the southern hemisphere. In the northern hemisphere, the zonal mean CH$_4$ shows strongly seasonal behavior as the maximum appears in summer and the minimum appears in spring. The overall increase trend in the northern hemisphere is larger than in the southern hemisphere and the gradient of the merge line between the northern and southern hemisphere has a tendency towards the southern. The corresponding increase rates of monthly zonal mean are derived from the EEMD decomposition analysis and are presented in Figure 7b. According to the increase rates, it is clearly shown that there is a large increase from the northern to the southern hemisphere around 2007 and there is evident phase delay of increase rates between the southern and the northern hemisphere. In the years afterwards, the increase rates over the tropics are slightly smaller than both the northern and southern subtropics regions. We can also find the slowing down of CH$_4$ increase around 2010 over the mid-to-high latitude region. The second rapid increase period starts around 2014. The increase rates reached to extremely high values over the subtropics to the high latitude region in the northern hemisphere and over the subtropics in the southern hemisphere, which are even larger than those in 2007. One small yet interesting feature is

from 2009 to 2016 as the increase rates of MUT-CH$_4$ over the tropics are slightly smaller than that over the northern and southern subtropics regions and the reasons are unknown.

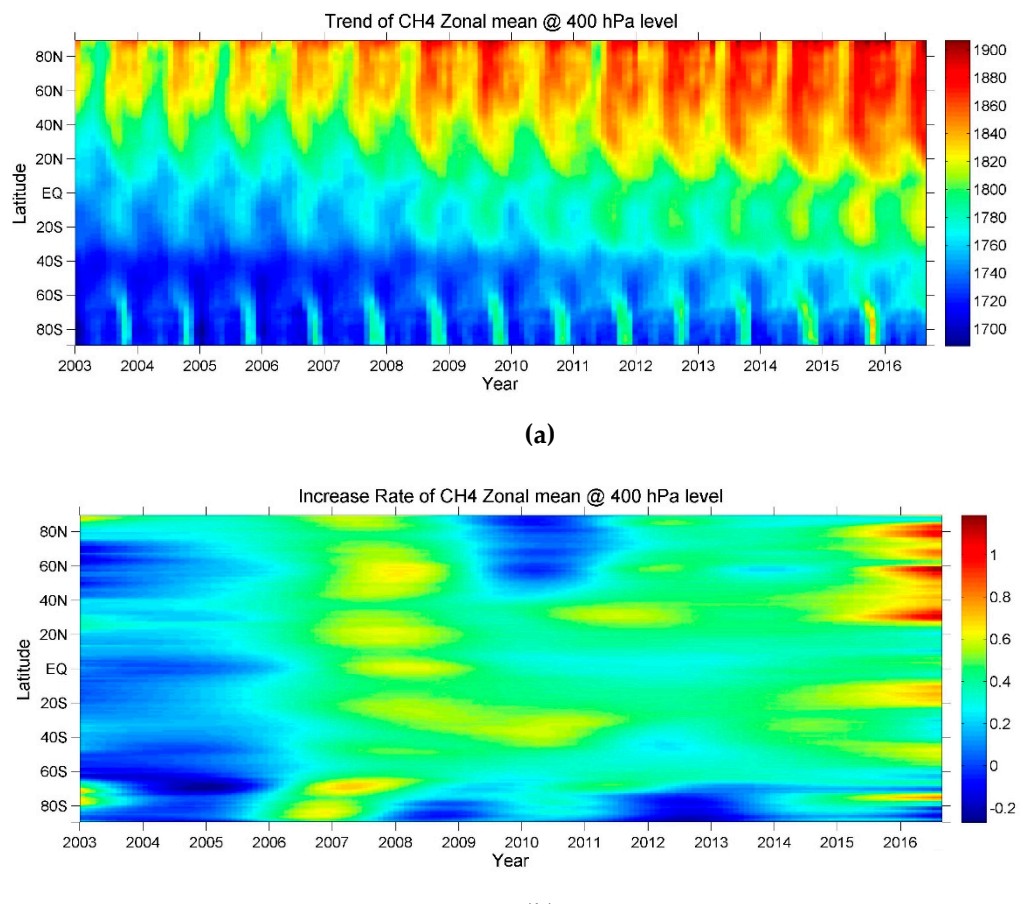

**(a)**

**(b)**

**Figure 7.** Zonal mean of CH$_4$ and increase rates (ppbv) in the mid–upper troposphere from AIRS. The zonal mean is calculated monthly from August 2002 to September 2016 with 1 degree latitudinal resolution. (**a**) monthly zonal mean of the mid–upper tropospheric (MUT) CH$_4$; (**b**) MUT CH$_4$ increase rates calculated using EEMD decomposed of monthly zonal mean.

In addition to the impact by emission sources, the CH$_4$ trend is also impacted by changes in the atmospheric CH$_4$ loss and soil uptake. Dalsøren et al. found that from 2000–2010, the modelled tropospheric OH column increased by 10–20% over China and India, and many model studies demonstrated a decrease in CH$_4$ lifetime due to an increase in global OH concentrations in the recent decades [51]. However, as discussed by Dalsøren et al., there is a missing consensus on OH trends among different studies. Some more studies also show a strong influence of El Niño–Southern Oscillation (ENSO) events on CH$_4$. Corbett et al. [52] also show the modulation of mid-tropospheric CH$_4$ by El Niño.

## 4. Conclusions

An advanced statistical analysis method, EEMD method, was used for the first time to analyze the trends of atmospheric CH$_4$ from 2003 to 2016 from three different measurements' datasets. The agreement of the annual increases in a few sites from this EEMD analysis with those from Bader et al. [4] proved the capability to use the EEMD method to analyze the CH$_4$ trends. Even more, by using EEMD analysis, we can derive the monthly increase rates of CH$_4$ which gives us more information about the CH$_4$ change with time.

Using this EEMD method, the monthly and annual increase rates of $CH_4$ in the marine boundary layer and the mid–upper troposphere, as well as the column average, were derived and compared in this study. Regardless of the difference in the uncertainties from these three measurements and their sensitivities to $CH_4$ variation in different altitudes, the analysis of the $CH_4$ trend and monthly/annual rates using three data sets in the northern hemisphere, tropics, and the southern hemisphere in this study demonstrated:

(1)   One common feature among these three measurements is that they have good agreement in capturing the abrupt increase of $CH_4$ in 2007. The increase rates of $CH_4$ in the MUT, as observed by AIRS, are overall smaller than $CH_4$ in MBL and the column-average $CH_4$.

(2)   During 2009–2011, there was a dip in the increase rate for $CH_4$ in MBL and the MUT-$CH_4$ increase rate was near zero in the mid–high northern latitude regions. The increase of the column-average $CH_4$, also reached a minimum correspondingly. Such a slow-down of increase rate might echo other studies that pointed out that the emission of $CH_4$ from industrial sources plays a large role in recent $CH_4$ increase as such a slow-down might be linked with the reduced industrial emission due to the financial crisis in 2008.

(3)   Increase rates reached another peak since 2014 which is even larger than in 2007, so a continual monitoring of the trends of $CH_4$ using both ground-based and space-borne measurement is important for climate change study.

**Author Contributions:** Conceptualization, M.Z. and X.X.; Data curation, M.Z., S.L. and Y.Z.; Formal analysis, M.Z. and X.X.; Funding acquisition, M.Z.; Investigation, X.X.; Methodology, Z.W.; Software, Z.W.; Supervision, X.X. and L.C.; Visualization, M.Z. and X.X.; Writing – original draft, M.Z. and X.X.; Writing – review & editing, M.Z. and X.X.

**Funding:** This research was funded by the National Natural Science Foundation of China, grant number 41771390, Program of Remote Sensing with high spatial resolution, 32-Y20A17-9001-15-17-2, and Project for data application of FY3 meteorological satellite.

**Acknowledgments:** We are grateful to the NOAA/ESRL/GMD for providing the ground-based measurements of $CH_4$ and the data used is from NOAA ESRL Global Monitoring Division, Carbon Cycle and Greenhouse Gases group, 325 Broadway R/CSD, Boulder, CO 80305 (http://esrl.noaa.gov/gmd/ccgg/). TCCON measurements from Park Falls were funded by NASA grants NNX14AI60G, NNX11AG01G, NAG5-12247, NNG05-GD07G, and NASA Orbiting Carbon Observatory Program. We are grateful to Jeff Ayers for technical support in Park Falls. The Lauder TCCON measurements were core-funded by NIWA through New Zealand's Ministry of Business, Innovation, and Employment. The TCCON $CH_4$ data is downloaded from https://tccon-wiki.caltech.edu/. The $X_{CH4}$ data from NDACC that was used in this publication were obtained as part of the Network for the Detection of Atmospheric Composition Change (NDACC) and are publicly available (see http://www.ndacc.org).

**Conflicts of Interest:** The authors declare no conflict of interest.

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
