# Peer review of "Increase of Atmospheric Methane Observed from Space-Borne and Ground-Based Measurements"

_remotesensing, doi:10.3390/rs11080964_

Round 1

Reviewer 1 Report

Review of the manuscript "Increase of Atmospheric Methane Observed from Space-borne and Ground-based Measurements" by Mingmin Zhou et al. submitted to Remote Sensing

The manuscript presents an analysis of global atmospheric methane concentrations based on satellite as well as ground-based in-situ and remote sensing measurements since 2002 to the present. As methane is one of the most important greenhouse gases which may considerably affect climate change and even global warming, the authors derive concentration trends using the EEMD statistical method. In my opinion, research goals set by the authors are very relevant for both scientific community and general public and the manuscript is of interest to Remote Sensing readers, however, a major revision of the paper is needed.

Major comments:

1. Paper structure

The structure of the paper is flawed. Instead of reviewing further sections of the paper in the last paragraph of the Introduction, the authors should use it to clearly state the aim of the paper, which is in the present version of the manuscript not evidently defined. The results according to these aims should be presented in Section 3 (Results and discussion) and the conclusions drawn in Section 4 (Conclusions). In my opinion, the summary in the Conclusions section does not bring any benefit to the manuscript and I would omit it.

2. Introduction

The introduction has many inconsistencies, for example in line 92-93 authors state that Only the AIRS data will be used for analysis of recent CH4 trend in this study as AIRS that has a data record from 2002 to present...” while in lines 106-110 they counter this by stating that “Since the sensitivity of AIRS is in the mid-to-upper troposphere, and the ground-based remote sensing measurements from TCCON and NDACC provide the column average, in this study we analyzed the trend of CH4 in the MBL from NOAA ground-based network, the trend of CH4 in the mid-upper, as well as the trend of the column-averaged CH4. Please clarify. Also, the details about the AIRS processing algorithm version are not relevant for the introduction, they belong into Section 2.

3. Citing

The authors should be consistent in citing other work. At present, they are mixing two citation styles, the Author (year) and the [number] style, and sometimes they even use both in a single citation case, which is superfluous and inadequate (for example Nisbt et al. (2016) [5] indicated). Thre are even cases when citations are written as “(see, e.g., Refs. 43 and 44)”. Please uniformize and use the [number] citation style only.

4. Tables

The data in the tables is hard to decode and it is sometimes not clear what is the benefit of presenting it in full in tabular form (trends are much easier seen from plots). The reader has to jump to and from in order to extract any data. As Table 1 defines regions for AIRS, it could as well define them for the whole planet so that all stations can be geographically categorized (no matter which of the three measurements they belong to). There is a lot of repetition between tables (for example 2 and 3). These two tables have inconsistent numbering of the sites and Table 4 appears to have a missing column before station name. Table captions in general and in particular those of Tables 5 and 6 should be more descriptive, and instead of these tables, figures would be preferred.

5. Results

Instead of presenting the results and emphasizing them with plots, the Results section reads as a description to a pictorial, the opening sentence being “Figure 3 shows...” (the same is true for subsection 3.2). The text of this section should present results, and refer to figures and tables for clarification, and not the other way around. Unfortunately, this is not the case and the results (trends, etc.) are not clearly stated in the text in a quantitative way. It is also very hard to follow a number of similar figures (Fig. 2 – 4). The authors would do much better to show one case as an example (like TR) and present the data for all regions together in a more compact way. The same is true for Figures 5 and 6.

6. Conclusions

This section should only include the conclusions based on measured facts. Summary should be excluded, as well as speculations such as slow-down might be linked with the reduced industrial emission due to the financial crisis in 2008”. This statement, which appears in various parts of the paper, is not supported by any facts provided by the authors nor by any citations, so it is a speculation and should be omitted (even if mathematical correlation does exist, this does not imply direct causality and needs detailed investigation).

The last section of the conclusions attempts to point out sources of uncertainties, which affect the retrieval of methane concentrations from different datasets. This is a very important issue that however does not belong in the conclusions. It needs to be addressed in more detail in Section 2 (it deserves its own subsection). The authors should estimate uncertainties of their results and provide error bars in concentration trend curves.

Minor comments:

L13: “stable density” → “stable concentration”

Everywhere: put “ “ between “(“ and previous word

L123: “in a 24-hour period, AIRS nominally observes the complete globe twice per day” →

in a 24-hour period, AIRS nominally observes the complete globe twice” or

AIRS nominally observes the complete globe twice per day”

L145: what is “ppbv”? It only appears here

L196: ftp url is not clickable

L201: Column 1 needs fixing

L316: Figure 6 should have its own caption and not “Same as Figure 5...”

Author Response

Dear Mr/Mrs,

Many thanks for the comments and suggestions helping us to improve this manuscript. Here we try to respond your comments.

 Major comments:

 1. Comment on Paper structure: The structure of the paper is flawed. Instead of reviewing further sections of the paper in the last paragraph of the Introduction, the authors should use it to clearly state the aim of the paper, which is in the present version of the manuscript not evidently defined. The results according to these aims should be presented in Section 3 (Results and discussion) and the conclusions drawn in Section 4 (Conclusions). In my opinion, the summary in the Conclusions section does not bring any benefit to the manuscript and I would omit it.

 We rewrite the last paragraph in the Introduction section. Goals of this study are emphasized. Also the Conclusion (Section 4) has been modified. Please see the revised manuscript for detail. The revised content has been marked with yellow background.

2. Comment on Introduction: The introduction has many inconsistencies, for example in line 92-93 authors state that “Only the AIRS data will be used for analysis of recent CH4 trend in this study as AIRS that has a data record from 2002 to present...” while in lines 106-110 they counter this by stating that “Since the sensitivity of AIRS is in the mid-to-upper troposphere, and the ground-based remote sensing measurements from TCCON and NDACC provide the column average, in this study we analyzed the trend of CH4 in the MBL from NOAA ground-based network, the trend of CH4 in the mid-upper, as well as the trend of the column-averaged CH4”. Please clarify. Also, the details about the AIRS processing algorithm version are not relevant for the introduction, they belong into Section 2.

The inconsistencies of descriptions on AIRS CH4 product have been corrected. And content on the AIRS processing algorithm version is removed from this section.

3. Comment on Citing: The authors should be consistent in citing other work. At present, they are mixing two citation styles, the Author (year) and the [number] style, and sometimes they even use both in a single citation case, which is superfluous and inadequate (for example “Nisbt et al. (2016) [5] indicated”). Thre are even cases when citations are written as “(see, e.g., Refs. 43 and 44)”. Please uniformize and use the [number] citation style only.

We checked the citation styles and use the citation style with reference to a sample paper which has been published on <Remote Sensing>.

4. Comment on Tables: The data in the tables is hard to decode and it is sometimes not clear what is the benefit of presenting it in full in tabular form (trends are much easier seen from plots). The reader has to jump to and from in order to extract any data. As Table 1 defines regions for AIRS, it could as well define them for the whole planet so that all stations can be geographically categorized (no matter which of the three measurements they belong to). There is a lot of repetition between tables (for example 2 and 3). These two tables have inconsistent numbering of the sites and Table 4 appears to have a missing column before station name. Table captions in general and in particular those of Tables 5 and 6 should be more descriptive, and instead of these tables, figures would be preferred.

We edited the Figure 1(b) by adding box to tag the region listed in Table.1. Table.2 and Table.3 listed ground stations for column-average and surface CH4 measurements respectively. According to the comment, we combine these original two tables into one and list the sequence of stations from north to south. Original Table.4 (now Table.3) is also modified. Table.5 and Table.6 are removed instead by plotted annual increase rate added to Figure 6 and Figure 7.

5. Comment on Results: Instead of presenting the results and emphasizing them with plots, the Results section reads as a description to a pictorial, the opening sentence being “Figure 3 shows...” (the same is true for subsection 3.2). The text of this section should present results, and refer to figures and tables for clarification, and not the other way around. Unfortunately, this is not the case and the results (trends, etc.) are not clearly stated in the text in a quantitative way. It is also very hard to follow a number of similar figures (Fig. 2 – 4). The authors would do much better to show one case as an example (like TR) and present the data for all regions together in a more compact way. The same is true for Figures 5 and 6.

We rewrite most parts in this Section. More description on the plots and discussion on the results with citations are added. We use Figure 2 as an example to show the components acquired from EEMD decomposition. More description on these components has been added. Figure 3 is intended to present the comparison between AIRS CH4 and NOAA surface CH4, while Figure 4 shows comparison between AIRS CH4 and column-average CH4 from TCCON. Unlike Figure 2, Figure 3 and Figure 4 are designed in a more compact way for two geo-matched pairs of CH4 measurements. Since AIRS mean CH4 over 5 regions and 7 in-situ CH4 measurements have been selected to make comparison of monthly/annual trend, it would be hard to decode if all these data was put in single Figure. So we divide these data into two groups according to the geographic location and plot the increase rates in Figure 5 and Figure 6 respectively.

6. Comment on Conclusions: This section should only include the conclusions based on measured facts. Summary should be excluded, as well as speculations such as “slow-down might be linked with the reduced industrial emission due to the financial crisis in 2008”. This statement, which appears in various parts of the paper, is not supported by any facts provided by the authors nor by any citations, so it is a speculation and should be omitted (even if mathematical correlation does exist, this does not imply direct causality and needs detailed investigation).

According to the comment, we have modified the Conclusion section. As to the speculation, discussion and citations have been added in Section 3.1.

The last section of the conclusions attempts to point out sources of uncertainties, which affect the retrieval of methane concentrations from different datasets. This is a very important issue that however does not belong in the conclusions. It needs to be addressed in more detail in Section 2 (it deserves its own subsection). The authors should estimate uncertainties of their results and provide error bars in concentration trend curves.

The last paragraph in the Conclusion introduces some key factors on remote sensing of CH4. They are vital to the accuracy of CH4 retrieval and should be considered when doing validation of the products between different observing systems. GFIT for total column retrieval and AIRS version 6 processing algorithm have been proved with good accuracy. In this study we focus on the trend analysis of CH4. Comparison of different observing systems should be in a separate paper. This part of content should not belong to the Conclusion. So we omit this part and add description on the uncertainty of EEMD method in Section 2.4.

Minor comments:

1.     L13: “stable density” → “stable concentration”

Everywhere: put “ “ between “(“ and previous word

Revisions according to the comments have been made.

2.     L123: “in a 24-hour period, AIRS nominally observes the complete globe twice per day” →

“in a 24-hour period, AIRS nominally observes the complete globe twice” or

“AIRS nominally observes the complete globe twice per day”

We rewrite this sentence as ‘in a 24-hour period, AIRS nominally observes the complete globe twice’.

3.     L145: what is “ppbv”? It only appears here

ppbv is unit of mixing ratio and it means parts per billion by volume. We add explanation in Line 136.

4.     L196: ftp url is not clickable

5.  L201: Column 1 needs fixing

We combine Table.2 and Table.3 into one. And this new table has several new columns.

6.  L316: Figure 6 should have its own caption and not “Same as Figure 5...”

We have edited Figure 5 and Figure 6. The captions of these two figures are also updated.

A copy of the revised manuscript with tracked changes is submitted. Please find it for detailed revision.

Best Regards,

Mingmin Zou and all co-authors.

Reviewer 2 Report

Monitoring of the trends of CH4 using ground-based and space-borne measurement is important for climate change study. In this study, the authors use a statistical method to analyze the trends of atmospheric methane (CH4) from 2003 to 2016 measured using an Atmospheric Infrared Sounder (AIRS), NOAA ground-based in situ measurements, and column-averaged CH4 in the atmosphere (XCH4) from the ground-based Fourier Transform Spectrometers at Total Carbon Column Observing Network (TCCON) and the Network for the Detection of Atmospheric Composition Change (NDACC). Using these datasets, the monthly and annual increase rates of CH4 in MBL, the mid-upper troposphere (MUT), and the column average were estimated. 

Results from the comparison of the CH4 trends in the mid-upper troposphere, lower troposphere and the column average from the three data sets showed an overall good agreement in capturing the abrupt CH4 increase in 2007 (1st peak) and an even faster increase after 2013 (2nd peak) over the globe. The rates of CH4 in the MUT were smaller than in the MBL and the column-average CH4. During 2009-2011 there was a decrease in the rate of CH4 in MBL. The increase of the column-average CH4 showed a minimum, thereby suggesting the trends of CH4 are not only impacted by the surface emission, but also by chemical reaction loss associated with [OH].

Author Response

Dear Mr/Mrs,

Many thanks for the comments and suggestions helping us to improve this manuscript. Here we try to respond your concerns.

 comments:

 Monitoring of the trends of CH4 using ground-based and space-borne measurement is important for climate change study. In this study, the authors use a statistical method to analyze the trends of atmospheric methane (CH4) from 2003 to 2016 measured using an Atmospheric Infrared Sounder (AIRS), NOAA ground-based in situ measurements, and column-averaged CH4 in the atmosphere (XCH4) from the ground-based Fourier Transform Spectrometers at Total Carbon Column Observing Network (TCCON) and the Network for the Detection of Atmospheric Composition Change (NDACC). Using these datasets, the monthly and annual increase rates of CH4 in MBL, the mid-upper troposphere (MUT), and the column average were estimated. 

Results from the comparison of the CH4 trends in the mid-upper troposphere, lower troposphere and the column average from the three data sets showed an overall good agreement in capturing the abrupt CH4 increase in 2007 (1st peak) and an even faster increase after 2013 (2nd peak) over the globe. The rates of CH4 in the MUT were smaller than in the MBL and the column-average CH4. During 2009-2011 there was a decrease in the rate of CH4 in MBL. The increase of the column-average CH4 showed a minimum, thereby suggesting the trends of CH4 are not only impacted by the surface emission, but also by chemical reaction loss associated with [OH].

 Major revision has been made to this manuscript. We revised the part of the Introduction to make the aim of this study more clear. More descriptions on EEMD method are added. Tables are edited and 2 of the tables are replaced with figures to show a better view of the data. We add more discussion on the EEMD decomposed CH4 trend and also revise the Conclusion.

.

A copy of the revised manuscript with tracked changes is submitted. Please find it for detailed revision.

Best Regards,

Mingmin Zou and all co-authors.

Reviewer 3 Report

The paper by Mingmin Zou et al. entitled “Increase of Atmospheric Methane Observed from Space-borne and Ground-based Measurements” is a study focusing on monthly and yearly increase rate of methane using dataset from the space-borne and ground-based measurements. Authors conducted data analysis using EEMD method to extract the increasing trend by removing periodic variation associated with the surface emission, transport and other processes. Based on the data analysis, they found that the increase rates of methane in the MUT are smaller than that in MBL and the column average values. Additionally, there is a dip in the increase of CH4 during 2009 – 2011, which might be linked with the reduced industrial emission due to the financial crisis in 2008. Furthermore, the increase rates reaches another peak since 2014. They concluded that EEMD analysis have an advantage to derive the monthly increase rate which shows the frequency of variability of CH4 increase rates in the mid-high northern latitude regions is larger than others.

Since methane is one of the most important greenhouse gases, this kind of analysis on the trend would be potentially important and would attract the researchers relating to the climate change. The first time implementation of the EEMD analysis on CH4 trend itself may also attract readers.

In general, method and results are reasonable. However, in my opinion, discussion part is not sufficient and thus conclusion is speculative. Furthermore, the strong point of this paper is not clear. More comparing with other papers on CH4 trend is required.

I have the following concerns.

Introduction, authors state that EEMD is implemented for the first time to analyze changing trend. However, why the application of EEMD is required is not stated. Authors should state what is the problems of conventional analytical method(s) and what is a merit expected to use EEMD.

Line 279-303, authors state that the increase trends have peaks at 2007 and 2014 whose main driver is anthropogenic sources in mid-latitude, and suggested that the financial crisis in 2008 cause the dip in 2009-2010. These are main conclusions of this paper. (conclusions (1), (2), and (3)). However, rapid increase at 2007 and 2014 was already reported and the reason of the increase from 2007 is a controversial issue. Many hypotheses, which involve changes in tropical wetland, live stock, fossil fuels, biomass burning and sinks, are proposed.(e.g. “interpreting contemporary trends in atmospheric methane” by Turner et al. PNAS 116 2805-2813 2019) Especially, isotopic studies suggest increase of tropical wetland and agricultural emissions as a major cause of methane increase from 2007. (e.g. “rising atmospheric methane: 2007-2014 growth and isotopic shift” by Nisbet et al. Global Biogeochem. Cycles 30 1356-1370 2016) Their conclusion is very different from that by authors. Therefore, authors should discuss about suggested causes other than anthropogenic emissions by Bader et al.

Line 301-302, if authors like to state financial crisis in 2008 as a cause of dip of increase, the clear evidence showing relationship between the crisis and industrial methane emission is required with reference at least.

Line 368-370, authors state that the increase rate of MUT-CH4 over the tropics are slightly smaller than both the norther and southern subtropics region as a conclusion. However, I could not follow the point of this statement. Please explain more why this result is important and with what this result is inconsistent.

Minor comments:

Abstract, although authors state about OH radical, it is not discussed in the text.

Figure 2, values on vertical axis are meaningless.

Table 4, remove “Unit: %/Yr”, instead replace “(%)” to “(%/Yr)”

Table 5 and 6 are shown but not used at all.

Figure 5 and 6, “ppbv” should be “ppbv/mo”

Author Response

Dear Mr/Mrs,

Many thanks for the comments and suggestions helping us to improve this manuscript. Here we try to respond your concerns.

 Major comments:

 1. Introduction, authors state that EEMD is implemented for the first time to analyze changing trend. However, why the application of EEMD is required is not stated. Authors should state what is the problems of conventional analytical method(s) and what is a merit expected to use EEMD.

 For any given time series from nature, the trend of it is a not known a priori. In such a case, using a straight line trend is a choice of none, since there is no physical justification that a trend should be linear. This awkwardness led to the statement that “one economist’s trend is another’s cycle” by Nobel Prize winner Engle and Granger. It was under such a situation that Wu et al. (Reference [44] in the revised paper) developed a completely new and logically consistent definition of the trend, which was a rare applied math paper published by PNAS. One of the key advantages of this EEMD derived trend is time varying, making the trend acceleration or deceleration a quantifiable quantity. Wu et al. (Reference [45] in the revised paper) has made comparison of global-mean surface temperature trends derived using EEMD and traditionally piece-wise linear fitting approach respectively. The featured multi-decadal variability from the linear fitted trend cannot describe the secular warming, while the warming rate of the EEMD trend is changing gradually, which provides a chance to diagnose the warming rate changes of the secular trend.

We have added more description on EEMD and its merit to traditional piece-wise linear fitting approach to Section 2.4 of this manuscript.

2. Line 279-303, authors state that the increase trends have peaks at 2007 and 2014 whose main driver is anthropogenic sources in mid-latitude, and suggested that the financial crisis in 2008 cause the dip in 2009-2010. These are main conclusions of this paper. These are main conclusions of this paper. (conclusions (1), (2), and (3)). However, rapid increase at 2007 and 2014 was already reported and the reason of the increase from 2007 is a controversial issue. Many hypotheses, which involve changes in tropical wetland, live stock, fossil fuels, biomass burning and sinks, are proposed.(e.g. “interpreting contemporary trends in atmospheric methane” by Turner et al. PNAS 116 2805-2813 2019) Especially, isotopic studies suggest increase of tropical wetland and agricultural emissions as a major cause of methane increase from 2007. (e.g. “rising atmospheric methane: 2007-2014 growth and isotopic shift” by Nisbet et al. Global Biogeochem. Cycles 30 1356-1370 2016) Their conclusion is very different from that by authors.  Therefore, authors should discuss about suggested causes other than anthropogenic emissions by Bader et al.

 Line 301-302, if authors like to state financial crisis in 2008 as a cause of dip of increase, the clear evidence showing relationship between the crisis and industrial methane emission is required with reference at least.

We add more discussion on the causes on CH4 trend. As well as the impact from emission sources, it is also influenced by atmospheric loss and soil uptake. Increase of tropospheric OH column may speed up the chemical loss of CH4. Many studies also show a strong influence of ENSO events on CH4 and the modulation of mid-tropospheric CH4 by El Niño. As to the influence of 2008 financial crisis, more studies on its impact to atmospheric trace gases are cited, which conclude the economic crisis from 2008 result in a reduction of gaseous pollutants, such as CO, NO2, SO2 and Volatile Organic Compounds (VOCs). GAINS-based model study also shows the economic crisis can lead to less GHG emissions using two economic projections developed before and after 2008 crisis. We have added all these discussions to Section 3.1 and 3.2.

While there are many hypotheses in studies on the main driver of CH4 trend, we hope the trends from our analysis would help the scientific communities to further explore the root-cause of the recent CH4 increase and to mitigate its potential impact to global warming.

3. Line 368-370, authors state that the increase rate of MUT-CH4 over the tropics are slightly smaller than both the norther and southern subtropics region as a conclusion. However, I could not follow the point of this statement. Please explain more why this result is important and with what this result is inconsistent.

We thought this looks interesting and might be related with some recent research about the change of tropospheric convections. It is really inclusive. But it is probably not a good idea to put it to the Conclusion. So we move it to Section 3.2 as a statement got from the Figure 7(b).

Minor comments:

1.     Abstract, although authors state about OH radical, it is not discussed in the text.

Discussion on OH radical has been added to Section 3.2.

2.     Figure 2, values on vertical axis are meaningless

Figure 2 shows an example of components decomposed using EEMD method. The left panel presents the residual Ri and the IMFs are shown in the right panel. To have a better view of Ri and IMFs of different order, scalars are added to Ri and IMFs when plotted. So the values on vertical axis indeed can’t represent the truth. They are set as a reference to show the scale of Ri and IMFs.

3.     Table 4, remove “Unit: %/Yr”, instead replace “(%)” to “(%/Yr)”

This Table has been modified and labeled as Table 3 in the revised manuscript.

4.     Table 5 and 6 are shown but not used at all

Table 5 and 6 has been replaced by renewed Figure 5 and 6.

5.  Figure 5 and 6, “ppbv” should be “ppbv/mo”

We have re-plotted Figure 5 and 6. The label of y-axis also have be modified.

A copy of the revised manuscript with tracked changes is submitted. Please find it for detailed revision.

Best Regards,

Mingmin Zou and all co-authors.

Round 2

Reviewer 1 Report

The revised manuscript and the attached comments/answers provided by the authors fully satisfy my concerns, so I recommend the manuscript to be accepted for publication after minor text editing. I suggest the authors to re-read the manuscript for typos and English language and style. A few examples of fixes that are in my opinion needed:

Typos and style

L27 – almost neglectable → almost negligible

L219 appendix of We et al [45]. → appendix of We et al. [45].

L413 It is noteworthy that the increase rates reach another peak →Increase rates reach another peak

Clarity
L209-212 – For any given time series from nature, the trend of it is a not known a priori. In such a case, using a straight line trend is a choice of none, since there is no physical justification that a trend should be linear. EEMD derived trend of a time series, a completely new and logically consistent definition of the trend, was developed [44]. One of the key advantages of this trend is time varying.

I understand what you mean, but please rephrase the above statements. Temporal evolution of quantities in nature which follow very well understood laws of physics (for example, trajectories of spacecrafts) can be (and actually are) calculated in advance. In the case of CH4 concentrations we are dealing with a complex system which we are not able to fully theoretically describe, hence, we do not know what the functional dependence of the temporal evolution of CH4 concentrations should be.

I suggest something like:

Temporal evolution of observables related to complex and not fully understood systems in nature is not a priori known.  Using the EEMD derived trend of the obtained CH4 concentration time series improves on the non-physical choice of simple linear dependence in a completely new and logically consistent way [44].

Author Response

Dear Mr/Mrs,

Many thanks for the comments and suggestions helping us to improve this manuscript. Here we try to respond your comments.

 Comments and Suggestions for Authors:

  The revised manuscript and the attached comments/answers provided by the authors fully satisfy my concerns, so I recommend the manuscript to be accepted for publication after minor text editing. I suggest the authors to re-read the manuscript for typos and English language and style. A few examples of fixes that are in my opinion needed:

 Following the suggestion, we have made some edition of the typos and English language through the whole paper.

Typos and style:

 L27 – almost neglectable → almost negligible;

L219 appendix of We et al [45]. → appendix of We et al. [45];

L413 It is noteworthy that the increase rates reach another peak →Increase rates reach another peak.

Clarity:

L209-212 – For any given time series from nature, the trend of it is a not known a priori. In such a case, using a straight line trend is a choice of none, since there is no physical justification that a trend should be linear. EEMD derived trend of a time series, a completely new and logically consistent definition of the trend, was developed [44]. One of the key advantages of this trend is time varying.

Revisions of typos and style have been made according to the comments. And We rewrite the sentence L209-212 following the suggestion.

A copy of the revised manuscript with tracked changes is submitted. Please find it for detail.

Best Regards,

Mingmin Zou and all co-authors.

Reviewer 3 Report

In this study, EEMD is implemented for the first time to analyze CH4 changing trendWhile authors just show the CH4 trend and leave further exploratory of the root-cause of the recent CH4 increase and mitigation of its potential impact to global warming to the scientific communities, the present approach may provide more information compared to traditional piece-wise linear fitting approach.

Author Response

Dear Mr/Mrs,

Many thanks for the comments and suggestions helping us to improve this manuscript. Here we try to respond your comments.

 Comments and Suggestions for Authors:

  In this study, EEMD is implemented for the first time to analyze CH4 changing trend. While authors just show the CH4 trend and leave further exploratory of the root-cause of the recent CH4 increase and mitigation of its potential impact to global warming to the scientific communities, the present approach may provide more information compared to traditional piece-wise linear fitting approach.

 Study on the reason of CH4 trend in past decades still remain many uncertainty. Further investigation of the root-cause may need more simulation from atmospheric model as well as observations. Here we try to show a picture of CH4 trend based-on EEMD method. Some editions on the typos have been made at round 2.

A copy of the revised manuscript with tracked changes is submitted. Please find it for detail.

Best Regards,

Mingmin Zou and all co-authors.
